# Elexacaftor/Tezacaftor/Ivacaftor Accelerates Wound Repair in Cystic Fibrosis Airway Epithelium

**DOI:** 10.3390/jpm12101577

**Published:** 2022-09-25

**Authors:** Onofrio Laselva, Massimo Conese

**Affiliations:** Department of Clinical and Experimental Medicine, University of Foggia, 71122 Foggia, Italy

**Keywords:** wound repair, airway epithelial cells, CFTR, cystic fibrosis, CFTR modulators

## Abstract

Background: Cystic fibrosis (CF) airway epithelium shows alterations in repair following damage. In vitro studies showed that lumacaftor/ivacaftor (Orkambi) may favor airway epithelial integrity in CF patients. Our aim was to evaluate the effect of the novel triple combination elexacaftor/tezacaftor/ivacaftor (ETI) on wound repair in CF airway epithelial cells. Methods: A tip-based scratch assay was employed to study wound repair in monolayers of CFBE14o- cells overexpressing the *F508del* mutation. ETI was added during wound repair. Results: ETI efficiently rescued CFTR F508del maturation and activity, accelerated wound closure and increased wound healing rates of the injured CF cell monolayers. Conclusions: The triple corrector/potentiator combination ETI shows promise in ameliorating wound healing of the airway epithelium in *F508del* patients.

## 1. Introduction

Cystic fibrosis (CF) is an autosomal recessive disease due to the occurrence of mutations in the CF transmembrane conductance regulator (CFTR) gene. To date, over 2000 different variants in the CFTR gene have been identified, although only about 400 are disease-causing [1], which have been subdivided into six different classes according to their functional ramifications [2]. The *F508del* mutation, the most prevalent mutation found in approximately 70–88% of the alleles in CF patients of Caucasian origin, belongs to Class II, thereby the CFTR protein is not appropriately transported to the plasma membrane. However, other functional defects associated with CFTR F508del concerning membrane stability and gating, have been found [3].

Since the introduction of CFTR modulators, a new era in the pharmacological treatment of CF patients has begun. Small-molecule drugs can be used as CFTR correctors, i.e., partially rescuing the trafficking defect produced by class II mutations, such as *F508del*, whereas others, called CFTR potentiators, are those that increase channel gating/conductance of CFTR proteins already positioned at the plasma membrane (class III and IV mutations) or after rescue to the cell surface (class II) [4]. While Orkambi^®^, a dual combination of the corrector VX-809 (lumacaftor) and the potentiator VX-770 (ivacaftor), led to an unsatisfactory amelioration of respiratory function and sweat chloride levels in *F508del* homozygous patients [5,6,7], a triple combination of two correctors—VX-661 (tezafactor) and VX-445 (elexacaftor)—with the potentiator ivacaftor, also known as Trikafta™, showed very high efficacy in patients having at least one *F508del* mutation [8,9].

Lung pathology in cystic fibrosis (CF) is represented in the overt disease by many alterations at the level of bronchi/bronchioli mucosa, such as hyperplasia of goblet and basal cells [10,11,12,13], squamous metaplasia [12,14], increase in epithelial height [11,13,15], cell shedding [10,11,12,15,16], and subepithelial fibrosis [10,17]. Although it is not completely clear which mechanism(s) underlie these alterations, a derangement in epithelium repair, regeneration and remodeling is likely to occur, and it is thought that CFTR lack/dysfunction plays a role in the dysregulation of the steps leading to a correct repair [18].

Some studies in primary CF airway epithelial monolayers have revealed that Orkambi^®^ treatment increased wound closure as compared to untreated controls and lumacaftor-only treated cells [19], and that Orkambi^®^ accelerated the repair of primary CF airway epithelial cells at the air-liquid interface in the absence and presence of *P. aeruginosa* exoproducts [19]. However, no studies to assess the enhancement of wound repair of CF monolayers in the presence of Trikafta™ have been performed.

Herein, we have explored if elexacaftor/tezacaftor/ivacaftor (ETI) could affect wound healing caused by mechanical injury of the CF airway epithelium by taking an immortalized CF bronchial epithelial cell line (CFBE overexpressing CFTR-F508del) as such a model. We found that F508del CFTR maturation and function were restored and wound repair accelerated in CF bronchial epithelial cells treated with ETI.

## 2. Materials and Methods

### 2.1. Cell Cultures

CFBE cells stably overexpressing F508del CFTR (CFBE-F508del) [20] were used. Cells were grown at 37 °C under 5% CO_2_ on flasks in MEM medium (Costar, Corning, MA, USA) containing 10% FBS, 1% L-glutamine, and 1% penicillin/streptomycin. The stably transfected cells were maintained in the presence of 2 μg/mL puromycin-positive selection.

### 2.2. Wound Repair Assay

CFBE-F508del monolayers were grown until confluency and injured mechanically with a P10 pipette tip [21]. A mark on the 24 well allowed us to photograph the wounds at exactly the same place at various times (at time 0 after injury and after 6 h, 24 h or 48 h). Digital images of 10x fields were obtained with a Leica DM IRB inverted microscope equipped with a Leica DFC450 C camera (Leica, Wetzlar, Germany). The percentage of wound closure and the wound healing rate, presented in μm/h, were calculated with ImageJ software (National Institutes of Health, Bethesda, MD, USA), as previously done [22].

### 2.3. Elexacaftor/Tezacaftor/Ivacaftor Treatment

CFBE-F508del cells were seeded in 24 well plates and were grown until confluency with MEM containing 10% FBS, 1% L-glutamine and 1% penicillin/streptomycin. Twenty-four hours before mechanical injury, cells were treated with the CFTR correctors VX-661 and VX-445 (both at 3 μM, from Selleck Chemicals, Houston, TX, USA) to allow F508del maturation. One hour before the mechanical injury, 1 μM VX-770 was added to the media containing the two CFTR correctors (VX-661 + VX-445, both at 3 μM) Fresh medium containing the triple drug combination (VX-661, VX-445 and VX-770) was replaced every 24 h post-wounding. In each experiment, two wells were analyzed per condition.

### 2.4. Evaluation of CFTR Function and Protein Maturation

#### 2.4.1. CFTR Channel Function in CFBE-F508del

CFBE-F508del cells were seeded in 96-well plates (Costar, Corning, MA, USA) and were grown at 37 °C for 5 days post-confluence as previously described [23]. Twenty-four hours before the fluorometric imaging plate reader (FLIPR) functional assay, the cells were treated with either 0.1% DMSO or 3 µM VX-661 + 3 µM VX-445. Cells were then loaded with blue FLIPR membrane potential dye (Molecular Devices, San Jose, CA, USA) dissolved in chloride free buffer (136 mM Na gluconate, 3 mM K gluconate, 10 mM glucose, 20 mM HEPES, pH 7.35) at a final concentration of 0.5 mg/mL. The plate was then read in a fluorescence plate reader (FilterMax F5, Molecular Devices, San Jose, CA, USA) at 37 °C. After 5 min baseline, F508del-CFTR was stimulated using 10 µM forskolin (FSK, Sigma-Aldrich, St. Louis, MO, USA) and 1 µM VX-770. After 10 min, a CFTR inhibitor (CFTRinh-172, 10 µM, Selleck Chemicals) was added to deactivate CFTR. The peak changes in fluorescence to CFTR agonists were normalized relative to the baseline fluorescence [24].

#### 2.4.2. Immunoblotting

CFBE-F508del cells were treated with 3 μM VX-445 + 3 μM VX-661 for 24 h at 37 °C. Then, the cells were lysed in modified radioimmunoprecipitation assay (RIPA) buffer (50 mM Tris-HCl, 150 mM NaCl, 1 mM EDTA, pH 7.4, 0.2% SDS, 0.1% Triton X-100) with protease inhibitor cocktail (Bio-Rad, Hercules, CA, USA) and analyzed by SDS-PAGE on 6% Tris-Glycine gels as previously done [25]. After electrophoresis, proteins were transferred to nitrocellulose membranes and incubated in 5% (*w*/*v*) milk in PBS 0.1% Tween 20. CFTR bands were detected using human CFTR-specific mAb 596 (1:2500 dilution, UNC, Chapel Hill, NC, USA) and Na/K ATPase α mAb (1:10,000 dilution, Santa Cruz Biotechnology, Dallas, TX, USA). The blots were developed with ECL (Bio-Rad), using the Chemidoc Imaging System (Bio-Rad) and the CFTR and Na/K ATPase α protein levels were quantified by densitometry using ImageJ.

### 2.5. Statistical Analysis

Statistical analysis was carried out using Prism for Windows, version 5.01, GraphPad Software Inc., San Diego, CA, USA. The Student t-test for unpaired data or ANOVA with Tukey’s post hoc test were used. Differences were considered significant when *p* < 0.05.

## 3. Results

### 3.1. Rescue of CFTR F508del Maturation and Activity by Elexacaftor/Tezacaftor/Ivacaftor

Recently it has been demonstrated that Trikafta™ treatment significantly rescued F508del-CFTR protein processing and channel activity in different cell line models and primary airway tissues [26,27]. In order to study the effect of this triple combination treatment on wound closure, we first validated the effect of Elexacaftor/Tezacaftor/Ivacaftor (ETI) on F508del-CFTR in CFBE-F508del cells. CFTR channel activity was measured using the membrane depolarization assay (FLIRP) and the protein expression by western blot.

ETI treatment of CFBE F508del-CFTR cells rescued F508del-CFTR activity as determined by the FLIPR assay (Figure 1A,B) as well as the maturation of CFTR protein as shown by the appearance of band C (Figure 1C,D).

### 3.2. Effect of Elexacaftor/Tezacaftor/Ivacaftor on Wound Closure and Wound Healing Rate

We have previously shown that CFBE cell monolayers (null for CFTR expression) show a delay in the wound closure. CFBE cells closed the wound only after 48 h, while non-CF 16HBE cells closed the wound in 24 h [22]. Thus, we investigated the effect of ETI on wound closure. As we previously showed, even CFBE overexpressing CFTR-F508del did not close the wound until 48 h (Figure 2a–d). Interestingly, the wound closure was accelerated by ETI treatment already at 6 h post-treatment (Figure 2f) and eventually the wound was almost entirely closed at 24 h (Figure 2g).

The evaluation of wound size (in % of the control, i.e., time 0) confirmed that by adding ETI to the injured epithelium, CFBE F508del-CFTR cells closed the wound almost entirely after 24 h (Figure 3A). Analysis of wound closure rates shows that the wound closure rate was significantly higher with ETI as compared with the untreated condition in the time interval of 0–6 h (Figure 3B). The wound healing rate decreased at 6–24 h for both experimental conditions, but it was still significantly higher for CFBE F508del-CFTR treated with ETI as compared with CFBE F508del-CFTR in the interval 0–24 h.

## 4. Discussion

Nowadays, the personalized treatment of CF patients is proceeding at an unprecedented pace due to the introduction of therapeutic drugs called CFTR modulators. These drugs can be used to treat nearly 90% of CF patients, including the CFTR potentiator ivacaftor, approved for residual function mutations (Classes III and IV), and combinations of correctors (lumacaftor, tezacaftor, elexacaftor) and ivacaftor for patients bearing at least one the F508del mutation [28]. To cover the other 10% of CF subjects and rarer mutations, other approaches based on novel CFTR modulators are being evaluated [29]. Based on clinical trials, CFTR modulators, in particular elexacaftor/tezacaftor/ivacaftor triple combination, have been shown to ameliorate sweat chloride concentrations and lung function [9]. Moreover, in vitro studies demonstrated that CFTR modulators, lumacaftor/ivacaftor, restore airway epithelial integrity in CF patients bearing class II mutation [19].

The defect in wound repair presented by us [22] and others in various cell models of CF primary and immortalized cell lines [19,21,30,31,32,33,34,35,36] strongly suggest that the lack/dysfunction of the CFTR protein may play an essential role. Orkambi^®^ (lumacaftor/ivacaftor) has been previously shown to significantly increase wound repair rates over a period of 6 h with CF primary airway epithelial cells [19]. Thus, it was of interest in the CFTR modulator era of personalized medicines, to assess that Trikafta™ exerted the same wound repair properties as Orkambi^®^. Of note, we observed that ETI treatment was able to increase the wound healing rate between 0 and 6 h, with a slowing of wound closure at further times, although still significant when the overall time interval 0–24 h was considered. These data suggested that rescuing F508del-CFTR by CFTR modulators increased wound repair approaching that of a non-CF cell line, i.e., 16HBE [22]. Since the triple combination of small molecules (VX-661, VX-445 and VX-770) directly interacts with CFTR and rescues the dysfunctional F508del-CFTR protein [37,38,39,40,41,42], the current study demonstrated that functional CFTR plays an important role for wound repair. Whether this is due to a boost to epithelial cell migration and/or proliferation [18] remains to be further evaluated. Moreover, the pathological microenvironment of CF airways is characterized by a complex interplay between pathogens and inflammatory/immune cells, governed by a wealth of bacterial products and cytokines/chemokines [43]. Our model is simplistic about these issues, therefore future studies will evaluate the recovery of wound closure by ETI in the presences of bacteria or their products together with inflammatory mediators. Finally, these studies will be further completed using primary airway respiratory epithelial cells, considered today a pre-clinical model for drug evaluation in the field of CF [44].

In conclusion, we have made the observation that treatment with ETI, in addition to rescuing the functional expression of F508del-CFTR, had an enhancing effect on the wound repair of a CF cell line overexpressing CFTR-F508del at earlier times. Future studies will be devoted to understanding the molecular cues determining these pro-restitution effects exerted by these drugs.

## Figures and Tables

**Figure 1 jpm-12-01577-f001:**
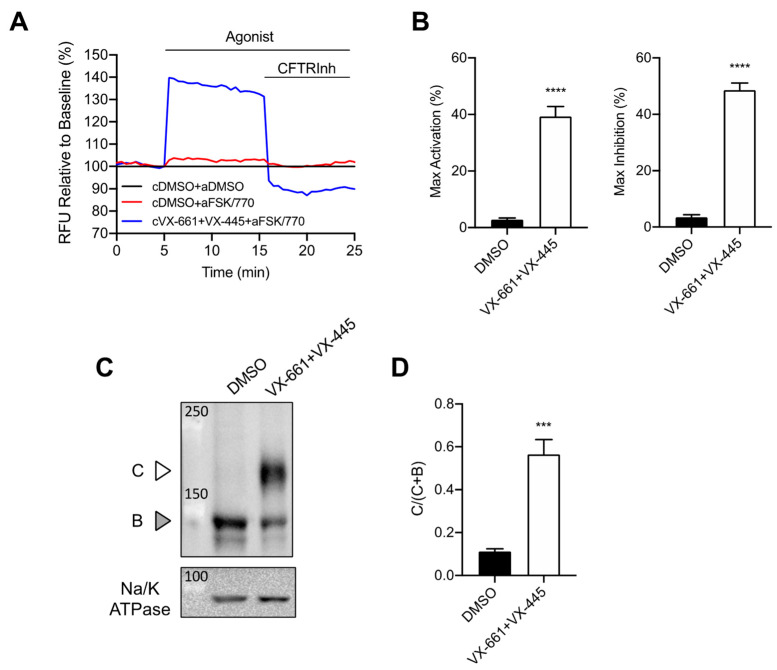
F508del-CFTR is rescued by VX-661 + VX-445 in CFBE F508del-CFTR cells. (**A**) Representative traces of F508del-CFTR dependent chloride efflux by membrane depolarization assay. CFBE-F508del cells were treated with 0.1% DMSO or 3 µM VX-661+ 3 µM VX-445 for 24 h at 37 °C. Acute DMSO for cells chronic pre-treated with DMSO (cDMSO + aDMSO), acute 10 μM forskolin (FSK) + 1 μm VX-770 (FSK/770) for cells chronic pre-treated with DMSO (cDMSO + aFSK/770), or acute FSK/770 for cells chronic pre-treated with VX-661 + VX-445 (cVX-661 + VX-445 + aFSK/770) were added as agonists. (**B**) Bar graphs show the mean (±SEM) of maximal activation of F508del-CFTR after stimulation by 10 µM FSK+ 1 µM VX-770 (left) or maximal inhibition by 10 µM CFTRInh172 (right) (*n* = 4 biological replicates with 4 technical replicates for each one). (**C**) Immunoblots of steady-state expression of F508del-CFTR in CFBE F508del-CFTR cells after 24 h treatment with 0.1% DMSO or 3 µM VX-661+ 3 µM VX-445. C: mature, complex-glycosylated CFTR; B: immature, core-glycosylated CFTR, Na/K ATPase as loading control. (**D**) Bar graphs represent the mean (±SEM) of the ratio band C/(band C + band B) (*n* = 4). *** *p* < 0.001, **** *p* < 0.0001.

**Figure 2 jpm-12-01577-f002:**
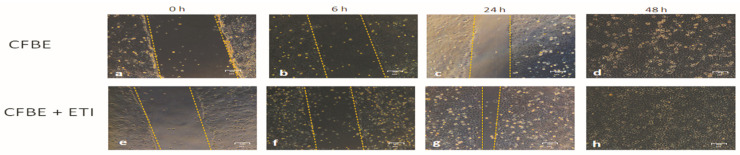
Wound closure in the presence of Elexacaftor/Tezacaftor/Ivacaftor (ETI) in CFBE F508del-CFTR cells. Wound repair was determined on CFBE F508del-CFTR (**a**–**d**), or CFBE F508del-CFTR with ETI treatment (**e**–**h**). Wound closure was evaluated at 0 h (**a**,**e**), 6 h (**b**,**f**), 24 h (**c**,**g**) and 48 h (**d**,**h**). Yellow dashed lines denote wound edges. Bar = 100 μm.

**Figure 3 jpm-12-01577-f003:**
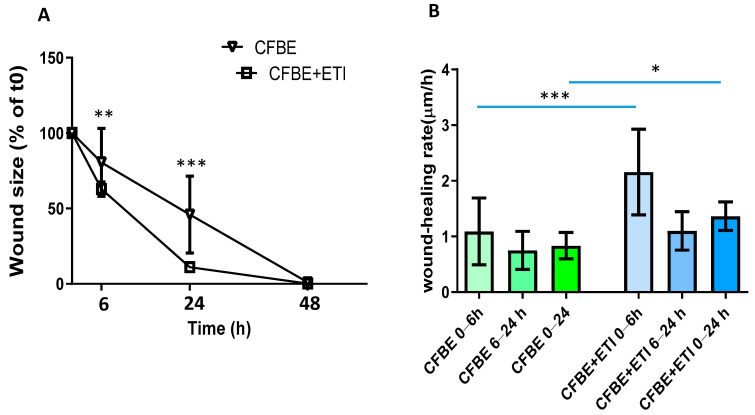
Wound size and wound healing rates in the presence of elexacaftor/tezacaftor/ivacaftor (ETI). (**A**) Percentage of wound size after 6, 24 and 48 h. Data are represented as a percentage of the initial area of the wound (t0) considered 100%. ** *p* < 0.01 and *** *p* < 0.001 CFBE-F508del + ETI vs. CFBE-F508del at 6 and 24 h, respectively (*n* = 3). (**B**) Comparison of wound closure rates post wounding (*n* = 3). * *p* < 0.05; *** *p* < 0.001. Data are shown as mean ± SD.

## Data Availability

The data that support the findings of this study are available from the corresponding author upon reasonable request.

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
