# Peer review of "Elexacaftor/Tezacaftor/Ivacaftor Accelerates Wound Repair in Cystic Fibrosis Airway Epithelium"

_jpm, 2022, doi:10.3390/jpm12101577_

Round 1

Reviewer 1 Report

The goal of the Laselva and Conese study by was to determine the impact of the combination elexacaftor/tezacaftor/ivacaftor (Trikafta TM) on wound healing in the cystic fibrosis airway epithelium. Using the CFBE-F508del cell line, they first validated the CFTR modulators by the analyse of their capacity to rescue the chloride currents through the F508del-CFTR and to improve the F508del-CFTR protein maturation. They next showed that incubation with VX445 + VX661 + VX770 combination accelerated the wound closure of CFBE-F508del cell monolayers. They conclude that the combination of CFTR modulators, in addition to rescue the functional expression of F508del-CFTR, had an enhancing effect on the wound repair of CF cells. 

The demonstration of a positive side effect of the combination elexacaftor/tezacaftor/ivacaftor is interesting and the results are convincing. However, I do have concerns with the study design and some clarifications are needed.

Major concerns

It is unclear why the authors used the CFBE-F508del cell line. Indeed, the authors reported previous results on wound closure delay that they obtained with the CFBE cell line (not overexpressing the F508del-CFTR).

The study would be improved by comparing CFBE cells (with empty vector), CFBE-WT cells (overexpressing the WT-CFTR) and CFBE-F508del cells (overexpressing the F508del-CFTR), all these cell lines having the same genetic background. In addition, the results deserve to be validated with primary cells from CF patients.

Minor concerns

Since TrikaftaTM is a registered trademark, I’m not sure if the authors have the right to use this name when using a combo of VX445, VX661 and VX770 compounds from Selleck Chemicals.

Immunoblot protocol should be more precise: the authors should specify that CF cells were treated with correctors/potentiators, the concentration of each modulator, the duration of the treatment before protein extraction, the amount of proteins to be separated, the concentration of tween ….

The cell treatment protocol prior to mechanical injury is unclear, especially lines 78-79. Do the authors mean that the culture medium was changed 24h after the wound with fresh medium containing the CFTR modulators?

Line 85: Are the cells treated with 0.1% DMSO and CFTR modulators? Isn’t rather: the cells are treated with either 0.1% DMSO or CFTR modulators?

Figure 1A: it would be better to write “FSK + VX770” instead of “agonists” above the curves, so as not to believe that the DMSO or the VX445/VX661 are put at the same time as the agonists.

Figure 1B: the authors should add the max inhibition too. 

Figure 1C: neither CFTR B and C bands nor protein names appear on the blots.

Figure 2 and Figure 3: It should be specified that cells are CFBE-F508del cells and not CFBE cells (in the figures and in the legends).

The text of the article should also be carefully checked as a whole to clarify this fact (figures 2 and 3, lines 140, 144,….) 

When the authors specified a n=3 or 4, does this mean that the experiments were carried out with 3 or 4 different passages of CFBE-F508del cells? How many wells were analysed per passage?

 One cannot speak of “rescue of expression” (lines 55, 80, 110) for the F508del-CFTR treated with correctors. Please replace “expression” by maturation for example.

Line 114: “validated” instead of “investigated”

Line 117: what do the authors mean by “determined an increase”? please reword.

Line 139: what do the authors mean by “injury was determined”? please reword.

The paragraph from line 143 to line 150 is difficult to understand and should be reworded.  

Abstract and line 167: “In this scenario, it is not known whether CFTR modulator therapies can operate a faster repair”; “Based on clinical trials, CFTR modulators have been shown to ameloriate sweat chloride concentrations and lung function [9], whereas other CF-associated basic defects, such as wound repair of the airway epithelium [30], have not been considered yet”.  

This is a bit exaggerated because the impact of Orkambi on epithelial wound repair has already been studied and published (reference 19).

Author Response

The goal of the Laselva and Conese study by was to determine the impact of the combination elexacaftor/tezacaftor/ivacaftor (Trikafta TM) on wound healing in the cystic fibrosis airway epithelium. Using the CFBE-F508del cell line, they first validated the CFTR modulators by the analyse of their capacity to rescue the chloride currents through the F508del-CFTR and to improve the F508del-CFTR protein maturation. They next showed that incubation with VX445 + VX661 + VX770 combination accelerated the wound closure of CFBE-F508del cell monolayers. They conclude that the combination of CFTR modulators, in addition to rescue the functional expression of F508del-CFTR, had an enhancing effect on the wound repair of CF cells.

The demonstration of a positive side effect of the combination elexacaftor/tezacaftor/ivacaftor is interesting and the results are convincing. However, I do have concerns with the study design and some clarifications are needed.

Major concerns

It is unclear why the authors used the CFBE-F508del cell line. Indeed, the authors reported previous results on wound closure delay that they obtained with the CFBE cell line (not overexpressing the F508del-CFTR).

A: As our reviewer mentioned, in the previous studies we performed experiments in CFBE cells because our aim was to evaluate human mesenchymal stem cells and fibroblasts as a therapeutic approach to reconstitute the airway epithelium integrity in CF patients. In the current work our studies focused on the effect of CFTR modulators on wound repair. Therefore, we used CFBE-F508del cell line because in CFBE cell line, which is null for CFTR expression (doi: 10.3390/ijms22105262), we couldn’t rescue any CFTR protein.

The study would be improved by comparing CFBE cells (with empty vector), CFBE-WT cells (overexpressing the WT-CFTR) and CFBE-F508del cells (overexpressing the F508del-CFTR), all these cell lines having the same genetic background. In addition, the results deserve to be validated with primary cells from CF patients.

A: The aim of the current work was to test the effect of CFTR modulators on wound repair in CF airway epithelium. Therefore, we chose to test only CFBE-F508del cell line because it is well demonstrated that CFTR modulators did not improve the maturation of WT-CFTR protein. However, previously studies already demonstrated by the directly comparison of CFBE-WT and CFBE-F508del-CFTR cells, an increased wound-healing rate of WT-CFTR compared to F508del-CFTR CFBE cells (doi: 10.1183/09031936.00221711).

We acknowledge the relevance of using primary airway epithelial cells in CF drug discovery and validation, whereas we wish to pinpoint that our study is a proof-of-concept appraisal of CFTR modulators on wound repair in CF airway epithelial cells.

Minor concerns

Since TrikaftaTM is a registered trademark, I’m not sure if the authors have the right to use this name when using a combo of VX445, VX661 and VX770 compounds from Selleck Chemicals.

A: We agree with our reviewer; therefore, in the revised version we used ETI (Elexacaftor/Tezacaftor/Ivacator) rather than Trikafta.

Immunoblot protocol should be more precise: the authors should specify that CF cells were treated with correctors/potentiators, the concentration of each modulator, the duration of the treatment before protein extraction, the amount of proteins to be separated, the concentration of tween ….

A: We now revised the method section by including all this technical information.

The cell treatment protocol prior to mechanical injury is unclear, especially lines 78-79. Do the authors mean that the culture medium was changed 24h after the wound with fresh medium containing the CFTR modulators?

A: We thank the reviewer and we now revised the method section.

Line 85: Are the cells treated with 0.1% DMSO and CFTR modulators? Isn’t rather: the cells are treated with either 0.1% DMSO or CFTR modulators?

A: We apologize for this misunderstanding. We now revised the sentence as the reviewer suggested.

Figure 1A: it would be better to write “FSK + VX770” instead of “agonists” above the curves, so as not to believe that the DMSO or the VX445/VX661 are put at the same time as the agonists.

A: We agree with our reviewer that was confusing, therefore we now revised the figure legend.

Figure 1B: the authors should add the max inhibition too.

A: We have now included the bar graph with the max inhibition (expressed as % of control, i.e. DMSO).

Figure 1C: neither CFTR B and C bands nor protein names appear on the blots.

A: We are sorry; probably it was a format problem. We now included a new figure with these labels on it.

Figure 2 and Figure 3: It should be specified that cells are CFBE-F508del cells and not CFBE cells (in the figures and in the legends).

The text of the article should also be carefully checked as a whole to clarify this fact (figures 2 and 3, lines 140, 144,….)

A: We now revised accordingly our reviewer’s comments.

When the authors specified a n=3 or 4, does this mean that the experiments were carried out with 3 or 4 different passages of CFBE-F508del cells? How many wells were analysed per passage?

A: The n=3 or 4 is not referred to 3 or 4 different passages of CFBE-F508del cells but to the number of experiments. In each experiment concerning wound healing, two wells were analysed per condition. We have added this information at the end of Section 2.3.

One cannot speak of “rescue of expression” (lines 55, 80, 110) for the F508del-CFTR treated with correctors. Please replace “expression” by maturation for example.

A: We have now corrected this miswording.

Line 114: “validated” instead of “investigated”

Line 117: what do the authors mean by “determined an increase”? please reword.

Line 139: what do the authors mean by “injury was determined”? please reword.

A: We have now changed and reworded as suggested.

The paragraph from line 143 to line 150 is difficult to understand and should be reworded. 

A: We have reworded this paragraph.

Abstract and line 167: “In this scenario, it is not known whether CFTR modulator therapies can operate a faster repair”; “Based on clinical trials, CFTR modulators have been shown to ameloriate sweat chloride concentrations and lung function [9], whereas other CF-associated basic defects, such as wound repair of the airway epithelium [30], have not been considered yet”. 

This is a bit exaggerated because the impact of Orkambi on epithelial wound repair has already been studied and published (reference 19).

A: We thank the reviewer and we now revised the Abstract and the cited paragraph. 

Reviewer 2 Report

This paper describes the effect of the triple combination CFTR modulator, TrikaftaTM on airway epithelial restitution. Scratch assays were performed using CFBE14o- cells overexpressing the F508del mutation to evaluate the effectiveness of therapy. The authors report that Trikafta rescued CFTR membrane expression/activity and accelerated wound closure rates in injured CF monolayers. The overall conclusion of the study was that Trikafta may effectively improve airway epithelial wound healing in F508del patients.  Overall, the results of the study support their conclusion but I have some questions and edits that need to be addressed.  

Major comments

1. Figure 1. The authors need to provide an explanation for the decrease in membrane potential below the 100% baseline value following treatment with CFTRinh172.

2. Figure 2. In order to eliminate the confounding issue of cell proliferation in the scratch assay, the authors should perform their experiments under serum free conditions. Removal of serum from the media for 24 hours prior to the experiment will move cells into Go and prevent them from proliferating during the assay. Under serum free conditions, the duration of wound closure will depend on the rate of cell migration.

3. Figure 1 and 3 legends: The authors need to include the number of times each experiment was performed (n = ?) for each part of the figures

Editorial Changes:

1.    Line 10: Replace “operate” with “produce”.

2.    Line 15: Replace “to ameliorate” with “in ameliorating”.

3.    Line 23: Replace “repercussion” with “ramifications”

4.    Line 31: Replace “are those increasing” with “are those that increase”.

5.    Line 32: Replace “in” with “of”.

6.    Line 52: Remove “the” before the word “wound” and replace “exerted” with “caused” and remove “a” that follows the word “by”.

7.    Lines 54-56: Awkward. Replace with “We found that F508del CFTR expression and function was restored and wound repair accelerated in CF bronchial epithelial cells treated with Trikafta”.

8.    Line 69: Replace “m/h” with “mm/h”.

9.    Line 70: Replace “as we previously did [22]” with “as previously described [22]”.

10.  Line 103: Replace “proteins” with “protein” and “level” with “levels”.

11.  Line 106: Remove “The” and capitalize “statistical”.

12.  Line 111: Remove “the” after “rescued”.

13.  Line 113: Replace “airways” with “airway”

14.  Line 115: Replace “cell line” with “cells”.

15.  Line 116: Remove “studies”.

16.  Line 117: Replace “determined” with “caused”

17.  Line 144: Replace “CFBE closed almost” with “closed the wound almost entirely after 24 h”.

18.  Line 145: Remove “the” after “while”

19.  Line 159: Remove “in the clinic of small molecules drugs” with “of therapeutic drugs called CFTR modulators”.

20.  Line 168: Remove “yet”.

21.  Line 169: Replace “cells” with “cell”.

22.  Line 172: Replace “increase significantly the” with “significantly increase”.

23.  Line176: Remove “already”.

24.  Line 179: Remove “the” after “increased”.

25.  Line 181: Replace “rescue” with “rescues”

26.  Line 182: Remove “channel”, replace “play” with “plays” and remove “the” after “for”.

27.  Line 192: Remove “the” after “that”, remove “the tropical combination” and replace “rescue” with “rescuing”.

28.  Line 195: Replace “understand” with “understanding”

29.  Line 196: Replace “pro-resolutive” with “pro-restitution”.

Author Response

This paper describes the effect of the triple combination CFTR modulator, TrikaftaTM on airway epithelial restitution. Scratch assays were performed using CFBE14o- cells overexpressing the F508del mutation to evaluate the effectiveness of therapy. The authors report that Trikafta rescued CFTR membrane expression/activity and accelerated wound closure rates in injured CF monolayers. The overall conclusion of the study was that Trikafta may effectively improve airway epithelial wound healing in F508del patients.  Overall, the results of the study support their conclusion but I have some questions and edits that need to be addressed. 

Major comments

  1. Figure 1. The authors need to provide an explanation for the decrease in membrane potential below the 100% baseline value following treatment with CFTRinh172.

 A: As previously studies demonstrated by our colleagues (i.e. Galietta, Lukacs, Pedemonte, Bear, Gentzsch; doi: 10.1172/jci.insight.139983; 10.3390/ijms23063175; 10.1183/13993003.01133-2018), often the CFTR inhibition values are higher than forskolin-dependent CFTR activation. Therefore, a lot CF scientist showed the CFTR inhibition responses rather than CFTR activation to maximize the CFTR modulator response. Therefore, we have now included the max CFTR inhibition in panel B of Figure 1.

Moreover, Eckford et al (doi: 10.1016/j.chembiol.2014.02.021) demonstrated that CFTR modulators, like VX-661, could induce the activation of F508del-CFTR protein previously rescued to the cells surface by CFTR modulators. Therefore, one explanation could be that the presence of CFTR correctors could exhibit a secondary activity by activating the basal phosphorylated F508del-CFTR after its partial rescue to the cell surface. Therefore, it can be concluded that the extra-inhibition by CFTRinh172 could be due to the partial basal activation of F508del-CFTR by CFTR modulators.

  1. Figure 2. In order to eliminate the confounding issue of cell proliferation in the scratch assay, the authors should perform their experiments under serum free conditions. Removal of serum from the media for 24 hours prior to the experiment will move cells into Go and prevent them from proliferating during the assay. Under serum free conditions, the duration of wound closure will depend on the rate of cell migration.

A: We do agree with our reviewer which makes a great point. However, we chose to be in a more relevant physiological condition, in the presence of serum, as previously published by Brochiero’s lab using airway epithelial cell cultures from CF patients (doi: 10.1016/j.jcf.2018.03.010; 10.1183/09031936.00221711).

  1. Figure 1 and 3 legends: The authors need to include the number of times each experiment was performed (n = ?) for each part of the figures

A: We are sorry for our mistake. We have now included the number of performed experiments where it was missing.

Editorial Changes:

  1. Line 10: Replace “operate” with “produce”.

  1. Line 15: Replace “to ameliorate” with “in ameliorating”.

  1. Line 23: Replace “repercussion” with “ramifications”

  1. Line 31: Replace “are those increasing” with “are those that increase”.

  1. Line 32: Replace “in” with “of”.

  1. Line 52: Remove “the” before the word “wound” and replace “exerted” with “caused” and remove “a” that follows the word “by”.

  1. Lines 54-56: Awkward. Replace with “We found that F508del CFTR expression and function was restored and wound repair accelerated in CF bronchial epithelial cells treated with Trikafta”.

  1. Line 69: Replace “m/h” with “mm/h”.

  1. Line 70: Replace “as we previously did [22]” with “as previously described [22]”.

  1. Line 103: Replace “proteins” with “protein” and “level” with “levels”.

  1. Line 106: Remove “The” and capitalize “statistical”.

  1. Line 111: Remove “the” after “rescued”.

  1. Line 113: Replace “airways” with “airway”

  1. Line 115: Replace “cell line” with “cells”.

  1. Line 116: Remove “studies”.

  1. Line 117: Replace “determined” with “caused”

  1. Line 144: Replace “CFBE closed almost” with “closed the wound almost entirely after 24 h”.

  1. Line 145: Remove “the” after “while”

  1. Line 159: Remove “in the clinic of small molecules drugs” with “of therapeutic drugs called CFTR modulators”.

  1. Line 168: Remove “yet”.

  1. Line 169: Replace “cells” with “cell”.

  1. Line 172: Replace “increase significantly the” with “significantly increase”.

  1. Line176: Remove “already”.

  1. Line 179: Remove “the” after “increased”.

  1. Line 181: Replace “rescue” with “rescues”

  1. Line 182: Remove “channel”, replace “play” with “plays” and remove “the” after “for”.

  1. Line 192: Remove “the” after “that”, remove “the tropical combination” and replace “rescue” with “rescuing”.

  1. Line 195: Replace “understand” with “understanding”

  1. Line 196: Replace “pro-resolutive” with “pro-restitution”.

A: We revised accordingly with the 29 suggestions from our reviewer.

Round 2

Reviewer 1 Report

The reviewer thanks the authors for the modifications.